# Quality Evaluation of New Types of Core Layers Based on Different Thicknesses of Veneers for Flooring Materials

**DOI:** 10.3390/ma17235881

**Published:** 2024-11-30

**Authors:** Sylwia Olenska, Piotr Beer

**Affiliations:** Institute of Wood Sciences and Furniture, Warsaw University of Life Sciences—SGGW, Nowoursynowska 159, 02-776 Warsaw, Poland; piotr_beer@sggw.edu.pl

**Keywords:** flooring, quality, composite, veneer, Shewart Control Charts, peeling

## Abstract

Problems with the availability of raw materials on the Polish market have forced wood industry producers to search for new, previously unused species of wood that meet the functional requirements of the target products. Therefore, it is necessary to conduct research on products whose structure is different from the popular offerings on the market. The goal of this study was to analyze the influence of the thickness of Scots pine veneers, also with Alder interlayer variants, on Young’s modulus and stiffness values of plywood-like composites dedicated to flooring applications regarding statistically based quality control of the products. The variables in this research are the thickness of the veneer, which creates the structure of the base layer of flooring materials, and the kind of wood used. This research looked at basic mechanical properties determining the suitability of flooring materials: modulus of elasticity and stiffness. Because both these parameters describe the product quality, the analyses were based on the normal distribution (containing kurtosis) and the creation of Shewart Control Charts for each parameter. Analyses of control charts provide information on whether the projected production process is stable and is able to give predictable results. In turn, the analysis of the kurtosis value allows us to determine whether Young’s modulus and stiffness values obtained for the products are as close as possible to the values assumed by the manufacturers. The thickness of veneers in the base layer of flooring composites can be enlarged, allowing production to be simplified and more environmentally friendly. New types of layered composites for flooring, manufactured by rotary cutting, without the need for quality assessment, with a minimum number of layers, and additionally verified with Shewart Control Charts, may be applied to production. Presented studies show that veneers of different quality classes, having plywood-like structures, can be used for flooring materials and that the thickness of the veneers in the base layer can be increased. In this way, wood can be used without the need for quality classification and with fewer production processes.

## 1. Introduction

Wood has been a useful material since ancient times. It is often used to produce furniture, floors, and other interior finishing elements. Everywhere it goes, it brings a note of charm and comfort. It is elegant and cozy at the same time. It fits into interiors arranged in various styles, from rustic, Provençal, Scandinavian, and boho to modern and glamorous. It is difficult to imagine a house without wooden furniture such as tables and chairs, wardrobes, chests of drawers, shelves, desks, display cabinets, or wooden floors [1].

Wooden floors have accompanied people in their everyday life since ancient times. Wood floors started to appear in homes about 400 years ago, replacing stone floors. The first mentions of wooden floors come from baroque France and colonial America; there, they were reserved for the manors and residences of the upper class. They were a manifestation of luxury and wealth. Wooden floors gained complete popularity in the 19th century. The first factories producing planks on a large scale were established during this period. Until World War II, parquet floors dominated, later replaced in part by linoleum, which was easier to install and more practical. Today, the fashion for wood in apartments is gaining strength again and is increasingly featured in the designs of decorators and stylists. This fashion is mainly dictated by the indisputable advantages that characterize various products made of solid wood, namely its universality and natural character, thermal insulation, and surface renewability [2].

While solid wood remains a material used for floors, wood composites began to be used for functional, economic, and environmental reasons. Various products have been introduced, such as the facing layer, made of wood with high aesthetic and mechanical properties, and the substructure layer, usually made of low-quality and inexpensive materials. Two composite structures can be distinguished. A two-layer structure, with a distinct face layer and substructure, and a three-layer structure, in which the substructure layer consists of an inner and a bottom layer [3,4]. The face layer is the top layer of the composite. Demands are placed on it regarding high aesthetic values and mechanical properties [5]. It is exposed to constant loads, e.g., furniture pressure, as well as cyclic loads, e.g., foot traffic. For this reason, high-density species are selected for the facing layer. The domestic species include oak (*Quercus* L.) and ash (*Fraxinus excelsior* L.), while among the exotic species are doussie (*Afzelia africana*) and sapele (*Entandrophragma cylindricum*). The base layer is made of one or more slats, with the thickness of the surface layer ranging from 3 mm to 6 mm. The inner layer is an essential element in layered composites. It determines the dimensional stability of composites as well as their mechanical properties. It is arranged perpendicular to the face and bottom layers. Elements of the inner layer may be made of lamellas. The lamellas connect the face and bottom layers with glue. There is no glue joint between the slats. Low-quality wood is used in the production of lamellas; most often, the wood species used are pine, spruce, or fir wood. The bottom layer is the last element of the composite. The main task of a three-layer structure is to eliminate the stresses of the facing layer. In a two-layer construction, the bottom layer is also the inner layer [6]. The bottom layer can be made of wood veneers with low mechanical properties. Loosely arranged lamellas connected lengthwise with micro-joints are also used for the bottom layer. The most commonly used woods for this are pine, spruce, and alder.

To reduce production costs and thus the prices of the final product, in addition to classic parquet and boards among wooden floors, manufacturers also offer panel-layered floorboards [7]. Produced from wood-like raw materials, they were created as an affordable alternative to parquet and boards while still maintaining the aesthetic values and mechanical properties of classic solid wood floors. Floor panels and boards are made of layers, usually two or three, and their surface may be laminated, wooden, or vinyl. However, the literature also discusses the potential disadvantages of using composite materials as flooring elements. The most frequently mentioned are [8]:Properties of many important composites are anisotropic—the properties differ depending on the direction in which they are measured;Many of the polymer-based composites are subject to attack by chemicals or solvents, just as the polymers themselves are susceptible to attack;Composite materials are generally expensive.Manufacturing methods for shaping composite materials are often slow and costly.

Various types of studies have been performed to explore how to avoid the significant influence of the cited disadvantages. The main topic of floor testing concerns the surface properties of the face, i.e., the outer layer of floors. Layered composites are made of solid or glued solid wood—as a two-lamella or three-lamella material. Hence, the tests are focused on testing the durability of the facing layer and are carried out only on the isolated material of the facing layer, for example, tests of hardness [9,10], strength [11], or tests after the modification of the face material [12]. Research into the physical and mechanical properties of floor composites for floating floors and sports floors is mainly conducted in terms of their elastic properties [13]. Impact studies between base and face layers are also being significantly undertaken on the impact of face-layer sliced lamella thickness and base type on surface-checking. The findings reveal an interaction between those layers towards optimizing the thickness of the top layer. It revealed low surface-checking for a standard solid wood lamellae base. It can be additionally improved by optimizing top-layer thickness [14]. As a face layer on the top of the flooring, in addition to the apparent visual effects, wood has a positive environmental influence [15] and a positive impact on one’s mental state [16]. Tests for other physical and mechanical properties are performed mainly at the request of flooring companies, though the results are not generally disclosed. Such research is conducted by faculties of universities [17] or institutes (e.g., the Institute of Wood Technology—ITD). Apart from research studies, international standards are established to equalize the properties of floors. One of the main ones is the parquet standard EN 13226:2009 [18] on wood flooring—solid parquet elements with grooves and/or tongues. It has been in force in Poland since 2004 and contains a list of wood species most often used for the production of wooden floors, a description of European and exotic wood species, along with the characteristics of their colors and shades, and a description of the effect of light after some time of exposure in unshaded places.

Working on optimal solutions for the production of goods and their use, as well as determining the conditions for what to do with them after their period of use, is currently the main topic and issue being dealt with. The product impacts and affects the environment during the whole production process, beginning with the material, through manufacture, and remaining up to the end-life of product waste [19]. Products that sustainably combine aspects help the economy change from linear to cascade and circular, help maintain natural resources, help save the environment, and, at the end of the chain, help people survive. The upcycling of wood and wood-based material waste, through the design of high-price sustainable materials, can help advance sustainable business by applying circular economy principles in the wood industry [20,21,22,23,24,25].

The goal of these studies was to analyze the influence of the thickness of Scots pine veneers, also with Alder interlayer variants, on Young’s modulus and stiffness values of plywood-like composites dedicated to flooring applications regarding a statistically based quality control of the products. The utilitarian goal of the research is to develop new flooring materials produced more efficiently from unusual wood species for this purpose whose structure is different from the popular offerings on the market.

## 2. Materials and Methods

### 2.1. Materials

Experimental samples, layered composites for flooring panels, were divided into four different groups by thickness and the material used for the base layers. Each group was made up of 10 samples. Base layers of industrial panels of three groups of experimental panels were made of A-class quality class Scots pine (*Pinus sylvestris* L.) in accordance with standard EN 1927-2:2008 [26]. The base layers of the fourth analyzed group were made of alder (*Alnus glutinosa* Gaerthn.) and Scots pine. The top layers of all groups were made of oak (*Quercus* L.). Figure 1 and Figure 2 present examples of veneers applied to manufacture the base layers of flooring materials.

The general construction of the flooring panels used for the experiments is presented in Figure 3.

The Scots pine and Alder veneers were cut by peeling (rotary cutting) to dimensions 350 mm × 180 mm. Veneers were cut on the industrial machine in Drewspan PPHU Skorupski-Wójcik SP.J. company (Wielopole Skrzyńskie, Poland) [28]. The layers in the samples were arranged as follows:-In even-numbered layers, the fibers were positioned perpendicular to the sample length (┴);-In odd layers, the fibers were positioned parallel to the sample length (=).

The described arrangement of layers in the samples, along with the thickness of individual layers for the 1st, 2nd, and 3rd groups of samples, are presented in Table 1.

In the case of the 1st, 2nd, and 3rd groups of samples, the results of their tests were compared with those of the industrial samples. The base layers of the industrial composites were achieved by sawing on a multi-blade sawing machine. The thickness of the tested and industrial base layers were the same. Figure 4 presents the structure of industrial samples.

The fourth sample group was made of floor panels made of pine and alder veneers. The pattern of layered samples and the thicknesses of individual veneer layers is presented in Table 2.

In the case of samples made using alder wood, the test results were compared with samples made of pine wood using the same veneer thickness, i.e., to Young’s modulus and stiffness results for Group 3.

### 2.2. Tests of Static Bending

The test of Young’s modulus (the modulus of elasticity (MOE)) was carried out on a TiraTest universal testing machine in a three-point scheme based on the standard EN 310: 1993 [29]. Figure 5 presents the standard scheme for testing the modulus of elasticity in the three-point test. The research materials and the testing method were presented [30].

The samples were positioned on supports with a spacing of 310 mm. The initial pressure force was 2 kN. The MOE in static bending was calculated on the basis of the standard formula:(1)$$Em=\frac{L_{1}^{3}(F_{2}-F_{1})}{4bt^{3}(a_{2}-a_{1})}$$
where the following is defined:

E_m_ [MPa]—modulus of elasticity;

L_1_ [mm]—distance between support centers;

F_2_ [N]—40% of maximum force;

F_1_ [N]—10% of maximum force;

b [mm]—sample width;

t [mm]—sample total thickness;

a_2_ − a_1_ [mm]—deflection arrow increment measured at the middle of the sample length (corresponding to F_2_ − F_1_).

One of the theories of determining stiffness is described in the ISO 9052-1 standard [31]. The stiffness of composites was calculated on the basis of the following formula:(2)$$k=\frac{Em*b*t^{3}}{12}$$
where the following are defined:

k [MNmm^2^]—bending stiffness;

E_m_ [MPa]—modulus of elasticity;

b [mm]—sample width;

t [mm]—sample total thickness.

### 2.3. Shewhart’s Theory Focused on the Subject of the Work

The analysis of the obtained test results for Young’s modulus and stiffness was carried out in three steps:Analysis of the normal distribution of results obtained for all experimental samples;Preparation of control charts for a group of industrial (comparative) samples and for experimental samples;Analysis of the control chart prepared for all experimental samples.

The tested samples were divided into several groups, and the analyses were first carried out for samples made only of pine layers but characterized by different thicknesses of individual layers. Then, the results obtained for the last group of samples were compared to the results obtained by alder-pine samples of the same thickness. Thanks to this, it was determined whether it is possible to use wood species other than pine in some of the composite layers while maintaining the physical and mechanical parameters.

In the first step, concerning the analysis of the normal distribution of the results obtained for experimental samples, the average value of the measurements, the Specification Limits, and the Control Limits were determined. Kurtosis was also determined, which is a measure of the number of extreme outliers.

When plotting the average values of the results obtained and the adopted Limits, all the above-mentioned elements are presented in the form of vertical lines spaced from each other by the designated values resulting from the calculated standard deviation. Specification Lines were calculated according to the following formula:(3)SL=Target±2δ
where the following are defined:

SL—Specification Line value;

Target—mean value for a group of tested samples;

δ—industrial samples standard deviation.

Control Lines were determined according to the following formula:(4)CL=Target±3δ
where the following are defined:

CL—Control Line value;

Target—mean value for a group of tested samples;

δ—industrial samples standard deviation.

The possible permissible deviation from the centerline in the process is determined by the designated Upper and Lower Control Lines. Upper and Lower Specification Lines are needed for stability control of the technological process. Many measurements between the Control and Specification Lines indicate the need to correct the process parameters.

The measurements must meet certain conditions to be analyzed with Shewhart Control Charts. First, the distribution of the values of the analyzed parameters must be a Gaussian normal distribution. Then, the Control Lines limit the area on the graph in which 99.7% of the values of this parameter lie [32]. If the presented condition is met, it is possible to analyze samples of various thickness groups using the Shewhart Control Chart.

Additionally, the kurtosis of the normal distribution for all experimental samples was analyzed. The value of kurtosis indicates how close a variable’s distribution is to a normal distribution [33,34]. It should be noted that kurtosis is defined in many sources as a measure of the “flattening” or “slenderness” of the distribution. However, this is an incorrect definition [35] because its value does not depend on what is happening “at the tip” (i.e., close to the central tendency) of the distribution but at its “tail”. So, in reality, kurtosis measures the occurrence of outliers.

The calculations used a standard formula for determining the value of kurtosis, in which its result was standardized to zero [36], i.e.,
(5)$$Kurt.=\frac{\frac{1}{n}\sum_{i=1}^{n}({x-M)}^{4}}{({\frac{1}{n}\sum_{i=1}^{n}{\left(x-M\right)}^{2})}^{2}}-3$$
where the following are defined:

n—number of observations;

x—value of a single observation;

M—mean of the results of the tested sample.

Depending on the obtained value of kurtosis, its nature was determined according to the following assumptions:Leptokurticity: kurtosis is greater than 0. The distribution has “heavier” tails compared to those in a normal distribution. This means that outliers are more likely to occur;Mesokurticity: kurtosis is equal to 0. The distribution has tails that are balanced compared to a normal distribution. This is a distribution whose kurtosis is the same as a normal distribution;Platykurticity: kurtosis is negative. The distribution has “lighter” tails than in a normal distribution. This means that the values cluster closer to the average value.

Four separate Shewhart Control Charts were created in the second step without setting normative values [37]. Individual measurements are compared to the average value of the tests performed for a given analyzed group. This is the most frequently chosen method in the case of analyses of numerically measurable values. The type of cards selected for the research analysis is standardized, which differs from the sequential cards proposed by various authors [38,39].

Similar to the normal distribution analysis, an essential element of standard Shewhart Control Charts is the counted lines that are drawn on the chart. The target corresponds to the expected value of the measured parameter. Its estimator is the sample group mean. Control Lines (Upper and Lower) are drawn parallel to the target, preceded by Specification Lines (Upper and Lower) at a distance [40].

The industrial samples determined the minimum values of Young’s modulus and stiffness that the experimental samples of all thicknesses should achieve. Based on these calculations, ranges for Control and Specification Lines were determined. For each new card, compression of the average value with the expected value for a given group was done. The received card had to be discarded if a significant difference between the compared values was detected.

In the last step, a Shewhart Control Chart was created for all experimental samples tested (separately for Young’s modulus and stiffness). In this case, the normative values were determined based on the measurement results obtained in the study (i.e., after 30 repetitions). On this basis, the value of the single analysis’s overall mean (target) and the mean value of the standard deviation were determined [41].

## 3. Results and Discussion

### 3.1. Analysis of Samples Made of Pine Wood

Following the assumptions of the Six Sigma method used for production, the produced groups of floor panels were measured. The measurements were taken to determine what was happening in terms of the product’s durability, especially from the customer’s perspective. The measurements performed are intended to ensure that the analysis and solution are based on actual results, not theoretical information. The measurement results and the calculated necessary indicators are presented in Table 3.

Figure 6 shows the results for the modulus of elasticity for the three groups of the tested experimental samples. It shows that the results fit the normal distribution. All the measured values are in the frame of Six Sigma (all samples are situated between Control Lines, and 29 of them are between Specification Lines). This allows us to proceed to Shewhart’s theory.

It should be noted, however, that in the case of a normal distribution of Young’s modulus values for pine-based samples, we are dealing with a platykurtic distribution. The calculated kurtosis value is −1.01, so the clustering coefficient is lower than 0. The obtained distribution is, therefore, characterized by a relatively large dispersion of the observed Young’s modulus numbers and a weak central tendency. The relatively slimmed-down shape of the graph is only due to the increased number of measurements appearing close to the nominal value. However, their number does not significantly affect the value of the concentration coefficient. Therefore, the obtained results show moderate difficulty differences and high interlocation.

Figure 7 presents the results for stiffness for all the tested experimental samples. It shows that the results fit the normal distribution. All the measurement results are in the frame of Six Sigma (all samples are located between Control Lines). This allows us to proceed to Shewhart’s theory.

Similarly to the analysis of the normal distribution of Young’s modulus, the kurtosis value was also calculated in the case of stiffness. In the case of a normal distribution for stiffness, the calculated kurtosis has a value of −0.37. Therefore, it is more than 30% higher than the kurtosis value for the normal distribution of Young’s modulus values. This proves that the normal distribution is significantly less flattened, and so the stiffness values obtained are much closer to the ideal normal distribution. Therefore, it can be concluded that, although the normal distribution of stiffness is also platykurtic, the values of the variable are much more concentrated around the mean compared to the value of Young’s modulus.

#### 3.1.1. Young’s Modulus

Table 4 presents the results of the calculated indices used to create Shewhart Control Charts of Young’s modulus for each sample group. The target was determined individually for each group of samples based on the average value of the measurements performed. The constant standard deviation value for industrial samples to analyze the dispersion of the parameter values was calculated. On this basis, Control Lines and Specification Lines were established.

The graphs presented in Figure 8 show the Shewart Control Charts for Young’s modulus results obtained for the industrial (comparative) samples and the three groups of samples made with different thicknesses of pine veneers. The results obtained for the experimental samples were compared to the indicators calculated for the industrial samples. In the case of the industrial samples, the analysis was performed based on three tested elements because, in the case of industrial production, the elements are characterized by very similar mechanical properties. At first glance, it is visible that the experimental samples show a much greater scatter of results (even in the individual analyzed groups) than the industrial samples tested. The observed large scatter of Young’s modulus results for the experimental samples also shows that not all the measurement values fall within the Control Limits established earlier for individual sample groups. However, in the case of all the analyzed groups, at least 60% of the Young’s modulus values are above the Lower Control Limit. Despite not being within the limits, values that fall above the Upper Control Limit are considered to be correctly made and have better properties than the remaining samples. This fact results from the very random selection of veneers of different quality classes to reduce production costs. At the same time, it is also noticeable that, despite the described scatter, even the experimental samples with the lowest Young’s modulus values reach higher values than the target assumed for industrial samples.

All the experimental samples were grouped into one Shewart Control Chart in the next step. The collected values are presented in Figure 9, which is based on the measurement values and calculations for Young’s modulus presented in Table 1. The total target of experimental samples is 8836 MPa, which is a higher value than the target count for the industrial samples. The difference between both values is approximately 16%. As already noted when analyzing the graphs in Figure 7, it is noticeable that all Young’s modulus values for the experimental samples are higher than the target set for the industrial samples. It is also essential that, in the case of analyzing all the groups of samples simultaneously, all the samples fall within the area of not only the Control Limits but also the Specification Limits. While it could be stated that this fact results from the increased scatter of results (in the case of their complete grouping), it still also shows the possibility of setting target process parameters to obtain repeatable and reproducible results.

#### 3.1.2. Stiffness

Table 5 presents the results of the coefficients calculations necessary to create the Shewart Control Cards for industrial samples and groups of samples made of pine veneers as elements of the base layers. One shared value of the standard deviation was adopted in order to maintain the comparability of the observed trends in all groups. The standard deviation was determined based on the results obtained for industrial samples, characterized by the highest repeatability and the smallest scatter of the obtained results.

Figure 10 shows the collected results calculated for stiffness concerning Shewhart’s theory for the industrial samples and the 1st, 2nd, and 3rd groups of samples. Similar to the analysis of Young’s modulus, the experimental samples show a large scatter of results. At the same time, all the observed stiffness values of the experimental samples are higher than in the industrial samples, indicating their good functional properties. The largest scatter of results was observed for the 3rd group of samples (samples with the most significant number of thinner layers), which also has the highest stiffness values. In this case, four results exceed the Control Limits. The highest stability and repeatability of results was observed for the 2nd group of samples, which in turn has the lowest average stiffness value. The obtained results allow us to state that all the tested experimental samples have similar stiffness values and similar trends in the scatter of results.

All the analyzed experimental samples showed higher stiffness values than the industrial samples. At the same time, due to the similar characteristics of the scatter of results of all the analyzed groups of pine samples, a basis was obtained for collecting all the results into one Control Card. In this card, one goal and shared values of Lower and Upper Limits were established. The results are presented in Figure 11, which is based on the measurement values and calculations for stiffness presented in Table 1. The total target of experimental samples is 330 MNmm^2^, which is a higher value than the target for the industrial samples. The difference between both values is approximately 25%. It is also essential that, in the case of analyzing all the groups of samples simultaneously, all the samples fall within the area of the Control Limits. This means that it is possible to produce industrial components with better mechanical properties than those currently available on the market. However, to implement industrial production, it is worth stabilizing the parameters of the production process, thus increasing the precision of the manufactured components.

### 3.2. Analysis of Samples Made of Pine and Alder Wood

#### 3.2.1. Young’s Modulus

Table 6 shows the results of Young’s modulus measurements performed for samples whose base layer was pine and alder veneers. In the subsequent stages of analysis, these results were compared with samples of panels based on Group 3 pine veneers, and the table also includes the results of the latter. Moreover, the kurtosis value was calculated for both groups to check the similarity/differences in the distribution of Young’s modulus values and total kurtosis to check the dispersion of results for all analyzed samples with the same thickness.

Similar to the analysis of samples based on pine layers, in the first stage, a graph of the normal distribution of Young’s modulus measurements was created for the group of samples made of alder and pine (shown in Figure 12). In the case of this group, rather a good fit of the distribution of measured samples to the theoretical normal distribution of the obtained values can be observed. At the same time, it should be noted that all the measured values of Young’s modulus are within the Six Sigma range, which makes it possible to perform further analyses using Shewart Control Charts.

The total kurtosis for the analyzed two groups of samples is −2.64. This indicates a significant flattening of the normal distribution of Young’s modulus values and, thus, a large scatter of the obtained results. It should be noted, however, that in the case of a normal distribution of Young’s modulus values for pine and alder samples, we are dealing with a platykurtic distribution. The calculated kurtosis value is −1.73, so the clustering coefficient is lower than 0. The obtained distribution, therefore, shows a relatively large dispersion of the observed values of Young’s modulus and a weak central tendency.

Consequently, the obtained results show moderate difficulty differences and high interlocation. It should be noted that the kurtosis value for samples made of pine wood is −1.71, which is very similar to the kurtosis value of pine alder samples. Therefore, an additional premise was found, indicating the possibility of comparing the Young’s modulus results of both groups of samples.

Table 7 shows the values of the targets, Control Limits, and Specification Limits calculated for the group of samples made only of pine (Group 3 from Section 3.1.1) and made of a combination of alder and pine layers. The standard deviation value of Young’s modulus has been preserved and is equal to the tolerance value assumed for industrial samples.

Figure 13 shows the Shewart Control Charts for the two groups of samples compared. When comparing samples made of pine and alder to samples made of pine, it can be seen that the former shows significantly higher values of Young’s modulus (average values differ by almost 14%). All the pine-alder samples also have higher values of Young’s modulus than the average value of Young’s modulus for pine samples. However, due to the large scatter of results observed in both groups of samples, it is necessary to stabilize the parameters of the future production process.

#### 3.2.2. Stiffness

The stiffness analysis for samples made of pine and alder was performed analogously to the analysis of Young’s modulus. Table 8 shows the stiffness values for the two groups of samples, with the kurtosis value calculated for each sample group and all of the analyzed samples. The table below maintains the order of the analyzed samples following Table 5. This makes it possible to analyze the mechanical properties of individual samples.

In the first stage, a graph of the normal distribution of stiffness measurements was created for the group of alder and pine samples (shown in Figure 14). In the case of this group, a good fit of the distribution of measured samples to the theoretical normal distribution of the obtained values can be observed. The total kurtosis for the analyzed two groups of samples is −0.88. According to the theory, the limit criterion of kurtosis is equal to |2| [42]. Kurtosis is within the range between −2 and 2, so it can be assumed that the stiffness measurements for the analyzed groups are free from outliers, which indicates that the distribution is close to the normal distribution. It should also be noted that all the analyzed kurtosis values (for Groups 3 and 4 and the common one) indicate a platykurtic distribution. The distribution has “lighter” tails than in a normal distribution. This means the values are clustered closer to the mean, and the tails are less developed. At the same time, it should be noted that all the measured stiffness values are within the Six Sigma range, making it possible to perform further analyses using Shewart Control Charts.

As in the case of Table 7, Table 9 shows the values of the targets, Control Limits and Specification Limits calculated for the group of samples made only of pine (Group 3 from Section 3.1.1) and the group made of a combination of alder and pine layers. The value of the standard deviation of stiffness has been preserved and is equal to the tolerance value assumed for the industrial samples.

Figure 15 compares the stiffness results for pine-alder samples and those made of pine only. Both groups displayed a similar average stiffness value, differing by less than 4%. It is noticeable, however, that the pine-alder samples showed far better stability of achieved stiffness values. Only one sample has a stiffness value exceeding the Control Limits, while in the case of pine samples, as many as four results are different from the Control Limits. Therefore, producing pine-alder elements will be easier to optimize in future output than producing pine elements.

Comparing the distribution of the obtained values for Young’s modulus and stiffness, it can be seen that the individual groups are characterized by the same tendency in terms of their increase and decrease. At the same time, it is noticeable that in the case of the analysis of Young’s modulus value, a more significant number of samples exceed the control limits than in the case of the analyzed stiffness. This is probably the effect of two factors. Firstly, the Control Limits were determined based on the results obtained for industrial samples, which were characterized by more excellent stability in the case of Young’s modulus. Secondly, it may be the effect of the thickness of the samples selected for the study, which is considered when determining the value of Young’s modulus and whose influence is leveled out when determining stiffness. This issue will be analyzed in future studies. The research presented in this article complements and extends the research on the influence of the quality of the raw material on the quality of layered composites for flooring materials [41].

## 4. Conclusions

Care for the natural environment forces the development of new products. Newly developed products must be manufactured, taking into consideration changes in base materials and environmentally friendly technologies. Changes in the method of using raw materials are primarily seeking to achieve their maximum use, with the minimum number of technological processes involved in making products from them. This task is fulfilled by the chipless cutting technology, which is rotary cutting (peeling). In the presented studies, it was shown that peeling fully replaces chip-cutting processes. In addition, it was shown that the number of veneers in the base layer can be reduced without changing the mechanical properties or even increasing them. This is particularly visible for the first group with the thickest layers, which is characterized by the smallest scatter among the experimental samples. The target values of Young’s modulus and stiffness increased compared to the industrial products. The need to optimize the technological process parameters causes the scatter of results for individual samples. In the case of the opening studies, this was not necessary. However, it is worth noting that the lowest values obtained for the experimental samples are still higher than for the industrial samples. Using alder veneer increased the mechanical properties by approximately 15%. These studies show that veneers of different quality classes, having plywood-like structures, can be used for flooring materials and that the thickness of the veneers in the base layer can be increased. In this way, wood can be used without the need for quality classification and with fewer production processes. This direction of new product development is entirely consistent with all the ideas and intentions of protecting the natural environment.

## Figures and Tables

**Figure 1 materials-17-05881-f001:**
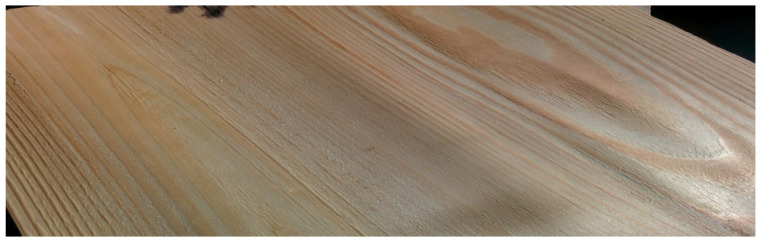
Pine veneer in the base layer.

**Figure 2 materials-17-05881-f002:**
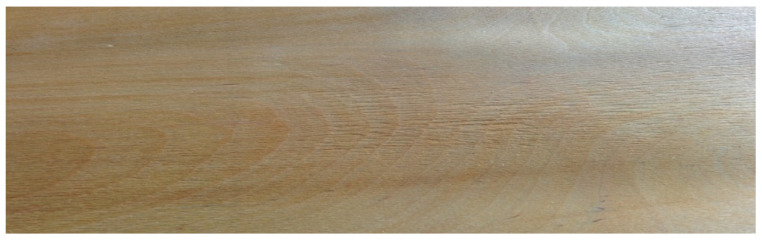
Alder veneer in the base layer.

**Figure 3 materials-17-05881-f003:**
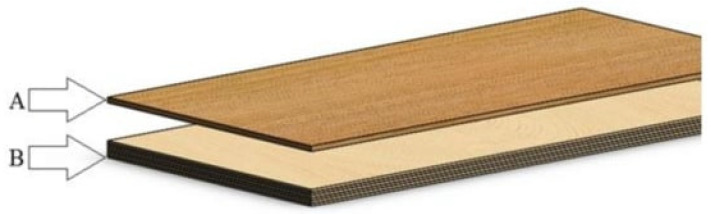
Structure of the composite with a two-layer structure: A—face layer, B—cross-shaped base layer [27].

**Figure 4 materials-17-05881-f004:**
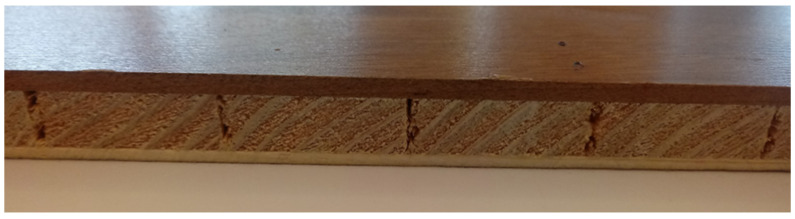
Structure of industrial samples.

**Figure 5 materials-17-05881-f005:**
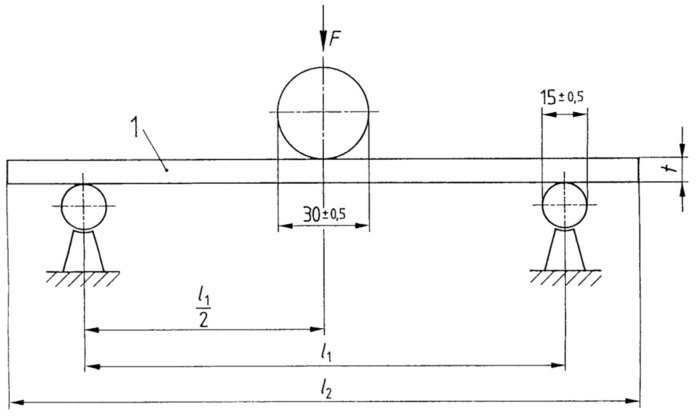
Arrangement of the bending apparatus: l_1_—310 mm, l_2_—340 mm, 1—sample, t—sample thickness.

**Figure 6 materials-17-05881-f006:**
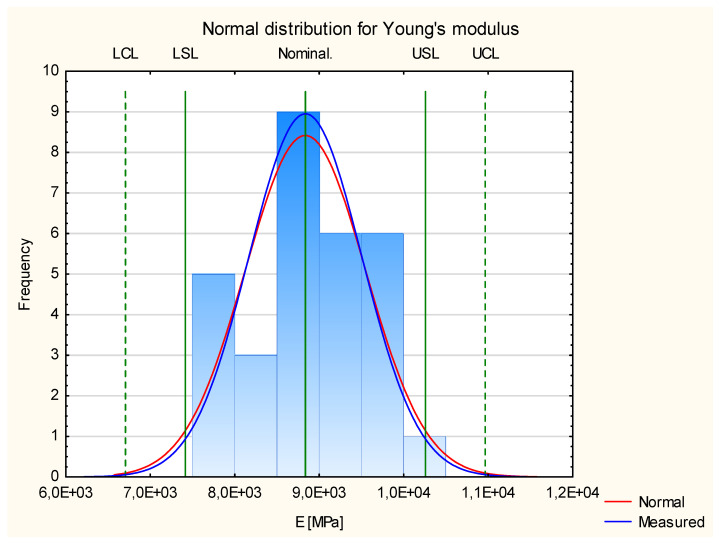
Normal distribution of Young’s modulus.

**Figure 7 materials-17-05881-f007:**
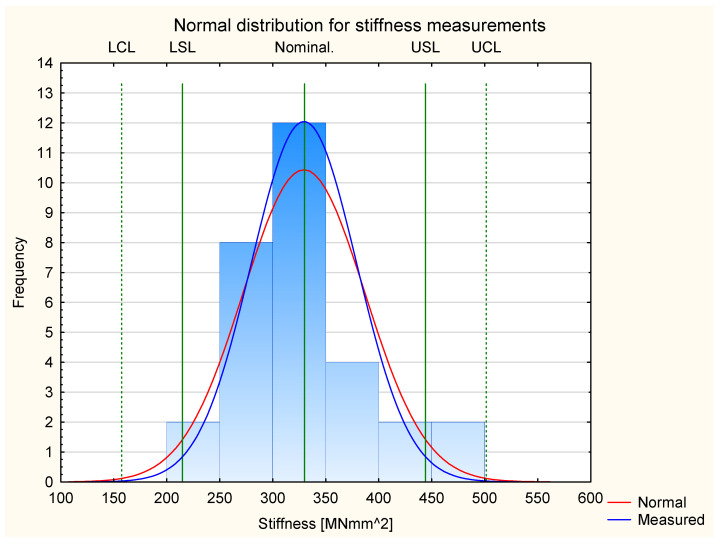
Normal distribution of stiffness.

**Figure 8 materials-17-05881-f008:**
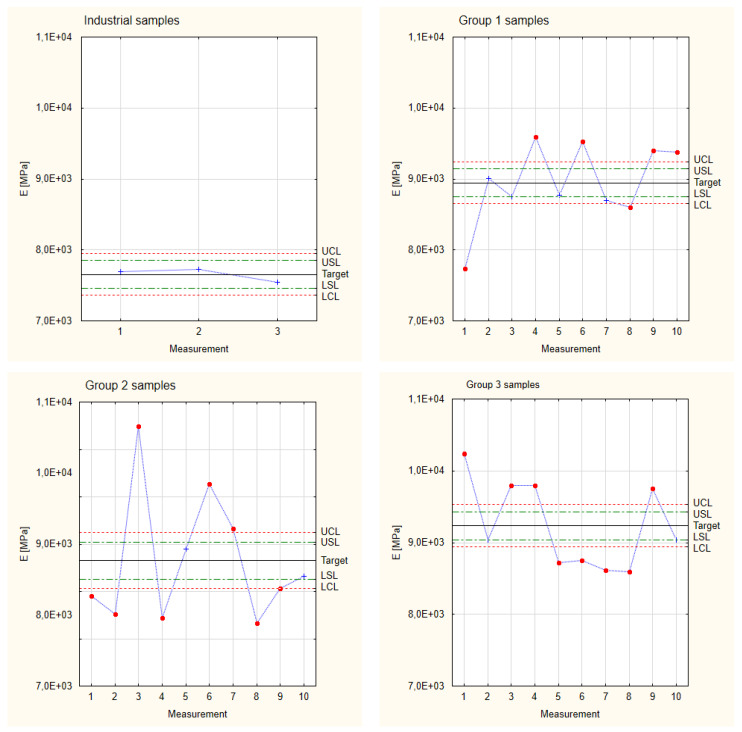
Shewhart Control Charts of Young’s modulus.

**Figure 9 materials-17-05881-f009:**
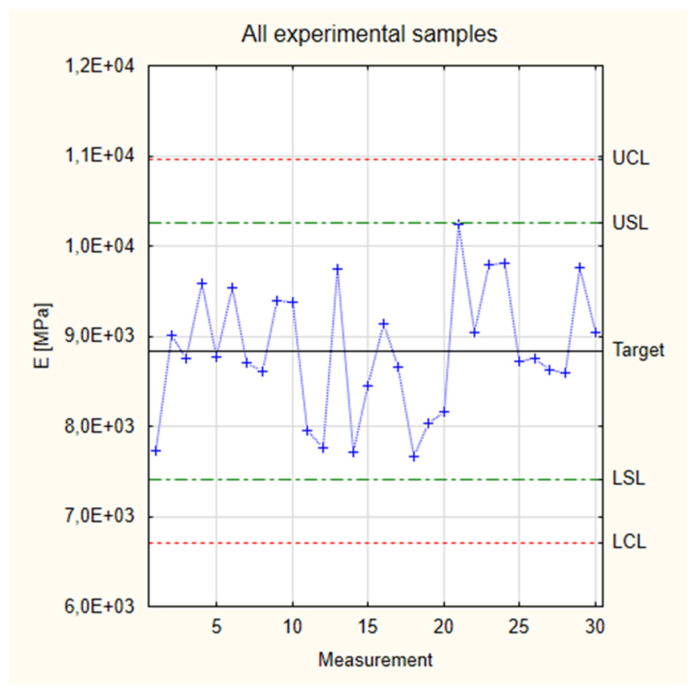
Collected test results of experimental samples for Young’s modulus.

**Figure 10 materials-17-05881-f010:**
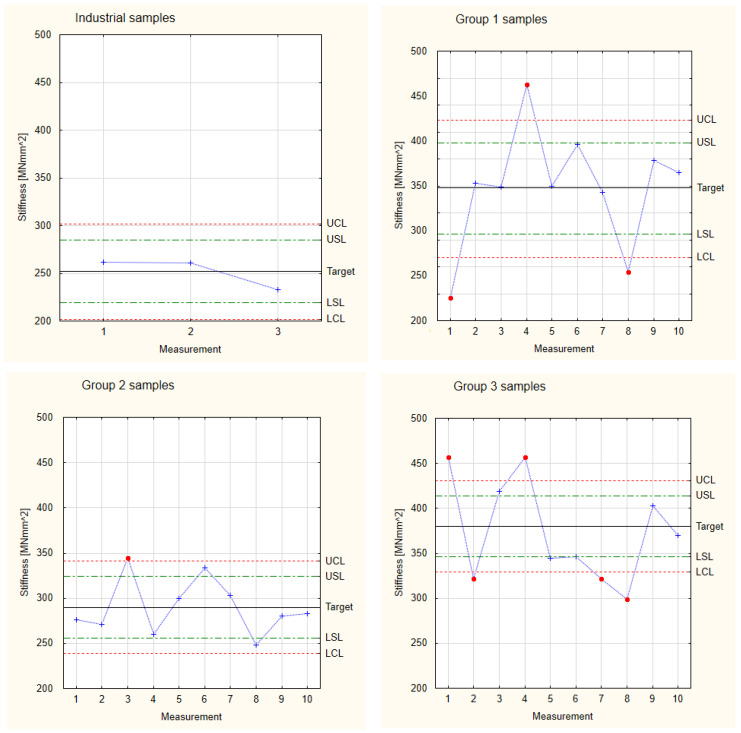
Shewhart Control Charts of stiffness.

**Figure 11 materials-17-05881-f011:**
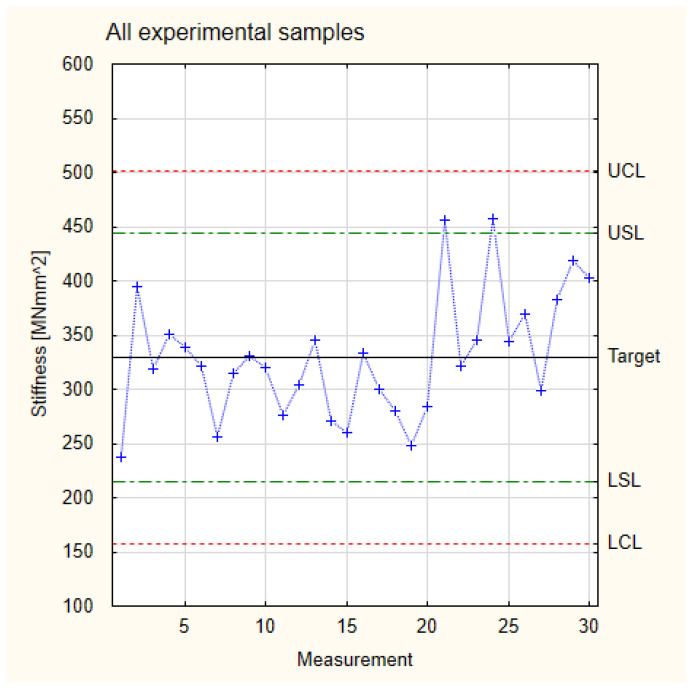
Collected test results of the experimental samples for stiffness.

**Figure 12 materials-17-05881-f012:**
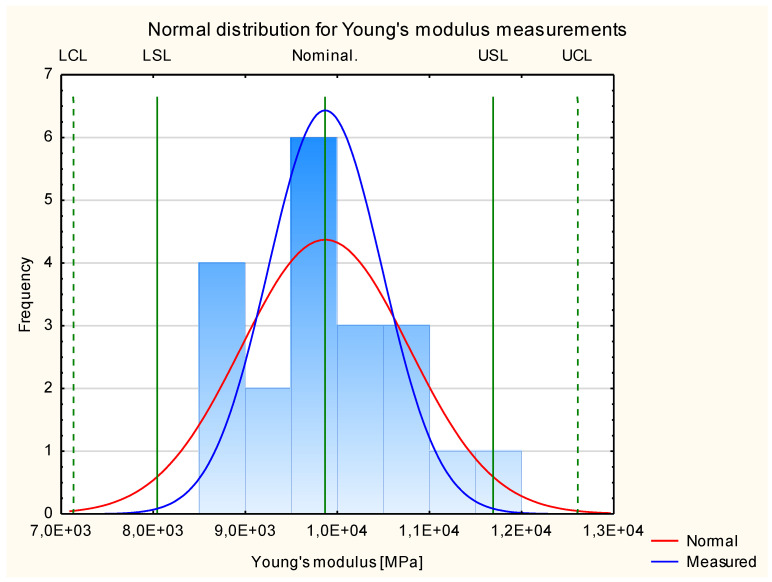
Normal distribution for Young’s modulus measurements of the 3rd and 4th groups.

**Figure 13 materials-17-05881-f013:**
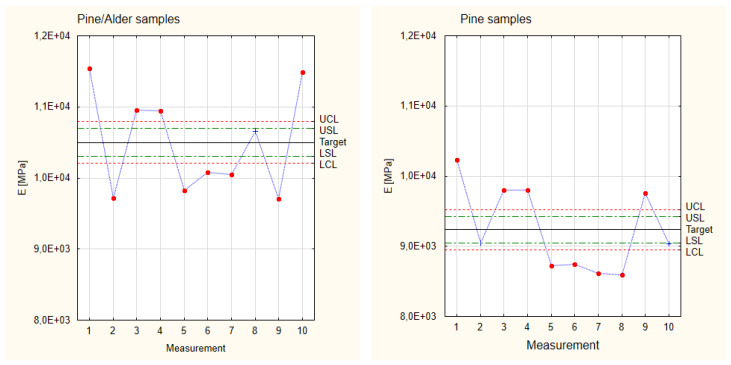
Shewhart Control Charts of Young’s modulus of the 3rd and 4th group.

**Figure 15 materials-17-05881-f015:**
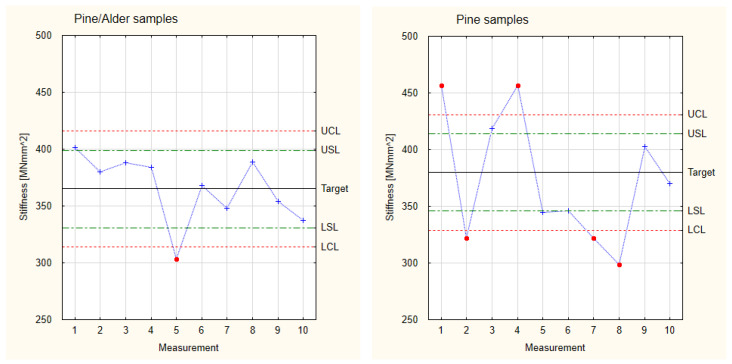
Shewhart Control Charts of stiffness of the 3rd and 4th groups.

**Figure 14 materials-17-05881-f014:**
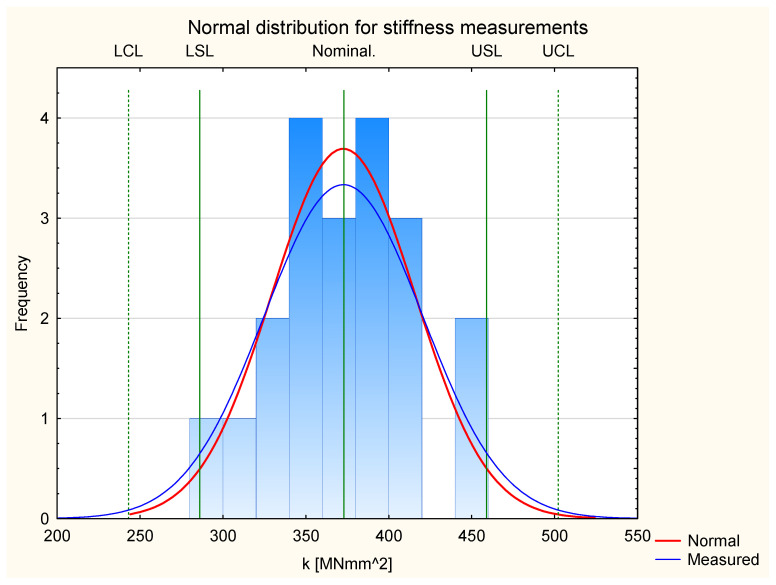
Normal distribution for stiffness measurements of Pine/Alder and Pine samples.

**Table 1 materials-17-05881-t001:** Scots Pine samples layout.

Sample Group	Oak Layer [mm]	Scots Pine Veneers in the Base Layer [mm]					
	=	┴	=	┴	=	┴	=
Group 1	3	2.5	3.2	2.5	3.2	-	-
Group 2	3	3.2	3.2	3.2	3.2	-	-
Group 3	3	1.5	2.5	1.5	2.5	1.5	2.5

**Table 2 materials-17-05881-t002:** Scots pine/alder sample layout.

Sample Group	Oak Layer [mm]	Alder Veneer [mm]	Pine Veneer [mm]	Alder Veneer [mm]	Pine Veneer [mm]	Alder Veneer [mm]	Pine Veneer [mm]
	=	┴	=	┴	=	┴	=
Group 4	3	1.5	2.5	1.5	2.5	1.5	2.5

**Table 3 materials-17-05881-t003:** Experimental test results.

Samples Group	Young’s Modulus [MPa]	Stiffness [MNmm^2^]
Group 1	7735	237
	9002	322
	8750	319
	9593	395
	8777	320
	9530	351
	8700	315
	8602	256
	9401	339
	9380	330
Group 2	7950	276
	7757	271
	9748	345
	7721	260
	8448	300
	9139	334
	8661	303
	7662	248
	8031	280
	8161	283
Group 3	10,236	457
	9043	322
	9800	419
	9802	457
	8726	345
	8750	346
	8618	322
	8594	299
	9755	403
	9036	370
Standard deviation	710	57
Lower Control Line	6704	158
Lower Specification Line	7415	215
Nominal (average value)	8836	330
Upper Specification Line	10,258	444
Upper Control Line	10,969	501

**Table 4 materials-17-05881-t004:** Values for Shewhart Control Chart of Young’s modulus values [MPa].

Sample Group	Standard Deviation	LCL	LSL	Target	USL	UCL
Industrial	98	7361	7459	7655	7851	7949
Group 1		8653	8751	8947	9143	9241
Group 2		8034	8132	8328	8524	8622
Group 3		8942	9040	9236	9432	9530

**Table 5 materials-17-05881-t005:** Shewhart Control Chart of stiffness values [MNmm^2^].

Sample Group	Standard Deviation	LCL	LSL	Nominal	USL	UCL
Industrial	17	202	219	252	285	302
Group 1		267	284	318	352	369
Group 2		239	256	290	324	341
Group 3		329	346	380	414	431

**Table 6 materials-17-05881-t006:** Experimental test results of Young’s modulus values.

Sample Number	Pine/Alder Samples [MPa]	Pine Samples [MPa]
1.	11,547	10,236
2.	9723	9043
3.	10,957	9800
4.	10,943	9802
5.	9830	8726
6.	10,076	8750
7.	10,050	8618
8.	10,660	8594
9.	11,492	9755
10.	9710	9036
Kurtosis	−1.73	−1.71
Total kurtosis	−2.64	

**Table 7 materials-17-05881-t007:** Values for Shewhart Control Chart of Young’s modulus [MPa].

Sample Group	Standard Deviation	LCL	LSL	Target	USL	UCL
Pine/Alder	98	10,205	10,303	10,499	10,695	10,793
Pine		8942	9040	9236	9432	9530

**Table 8 materials-17-05881-t008:** Experimental test results of stiffness values.

Sample Number	Pine/Alder Samples [MPa]	Pine Samples [MPa]
1.	401	457
2.	380	322
3.	388	419
4.	384	457
5.	303	345
6.	368	346
7.	348	322
8.	389	299
9.	354	403
10.	337	370
Kurtosis	−0.71	−1.49
Total kurtosis	−0.88	

**Table 9 materials-17-05881-t009:** Values for Shewhart Control Chart of stiffness values [MNmm^2^].

Sample Group	Standard Deviation	LCL	LSL	Target	USL	UCL
Pine/Alder	17	314	331	365	399	416
Pine		329	346	380	414	431

## Data Availability

The original contributions presented in this study are included in the article. Further inquiries can be directed to the corresponding author.

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
