# Peer review of "Quality Evaluation of New Types of Core Layers Based on Different Thicknesses of Veneers for Flooring Materials"

_materials, 2024, doi:10.3390/ma17235881_

Round 1
Reviewer 1 Report
Comments and Suggestions for Authors
The topic of the research work and manuscript is really interesting and provides new information. However there are several issues to be addressed towards its quality improvement before publication. Pay attention to the sub-/super-scripts that are not in the appropriate format. The scientific species names should be in italics. In line 67, please provide here a reference. In line 99, please provide more information. In the case of standard EN 13226:2004, please check it again, since I believe it has been withdrawn and check all the standards for their year of publishing or updated. In line 116, please add the relevant work of https://link.springer.com/article/10.1007/s00107-024-02114-x#citeas:~:text=DOI-,https%3A//doi.org/10.1007/s00107%2D024%2D02114%2Dx,-Share%20this%20article as a reference. You should clarify on the type of veneers used (rotary or sliced) from the first time mentioned in materials-methods section. Provide more information concerning the raw materials/elements used in the panel production. which was the equipment used for the rotation cut?Provide the DOI numbers. The statistical analysis provided is not very clearly presented in the respective section, please provide more information if possible. In conclusions section start, something is missing. Chech the first sentence. In line 570, please replace with the symbol %.
Comments on the Quality of English LanguageAcceptable. In line 107, please correct the "has been".
Author Response
Dear Reviewer, thank you for comments. We have corrected our article according to them. Only decision about the DOI numbers we less for Editor.
Reviewer 2 Report
Comments and Suggestions for Authors
The manuscript describes and discusses an experimental campaign on wooden floor boards composed of different veneer numbers and layouts. Some aspects of the adopted methodology and results would benefit from additional explanation and clarification; I would recommend the Authors to carefully check and amend the manuscript in this regard. Some relevant comments are reported below:
Line 2: should not it be “new types of core layers”?
Line 11: please correct the typos, it is Pinus sylvestris and plywood-like composites
Line 14: Instead of cheaper, I would suggest to prefer adjectives such as affordable, inexpensive, cost-effective, economical, etc., throughout the manuscript
Lines 59-60: please adopt italic when reporting the botanical name of the wood species
Lines 95-97: please check this sentence once more: is there any verb or term that is missing?
Line 97: “this study” here seems to refer to a literature source, but might be also meant as the work you are presenting. Please check and rephrase.
Line 112: Is not “production of products” a bit redundant?
Lines 137-138: I would suggest to also mention the finishing layer made of oak here, since it is then reported in Tables 1-2 only. Furthermore, later these specimens are compared to industrial specimens, for which in lines 167-168 it is mentioned that thicknesses of layers are the same: how are the industrial samples different? Is it only because of poorer material properties, or orientation of layers, or other factors?
Line 197: please check that “L” should be “L1” according to Equation 1
Line 201: to avoid any misunderstanding, I would suggest to specify that t in Equations 1 and 2 is the total thickness of the samples
Line 213: I would recommend to specify that k is the bending stiffness in Equation 2
Line 216: the correct unit is MN/mm2
Line 228: it should be “analyses”
Line 266: Please check the term “card” here and in several other sentences in the manuscript, as I expect it should be substituted with “chart”
Line 295: I am more familiar with the spellings “platykurticity” and “platykurtic”, thus I suggest to check whether the spelling in the manuscript is also applicable.
Line 329, Table 1: the reported values of stiffness seem not to be correlated with the corresponding Young’s moduli: for example, in Group 1, a value of 9002 MPa is associated to a stiffness of 395 MN/mm2, but a value of 9593 to only 351 MN/mm2. Equation 2 only depends on geometry and Young’s modulus, so there should be correlation: were there geometry variations (thickness, widths, …) in the samples? If so, it is important to mention them. Otherwise, please check the obtained values and provide a relevant explanations.
Line 490, Figure 11: please change one of the two colours in the graph, similarly to Figures 5 and 6, and also check that the legend correctly reports the Pine/Alder and Pine only samples. The same applies to Figure 13.
Line 554, Figure 14: how do the authors explain the fact that the stiffness for the pine/alder samples show a much better fit within the control limits compared to the modulus of elasticity of the same samples? The stiffness directly depends on the modulus of elasticity, so the dispersion should also be similar.
Line 557: please pay attention to the fact that the Conclusions are not starting with a complete sentence.
Line 561-563: the chipless cutting is here mentioned for the first time. More background on this has to be reported in the introduction or in section 2. How does this also compare to the industrial samples?
Line 564: does the statement on reduced veneers refer to the properties of e.g. group 1 samples, having a lower amount of veneers, compared to the others? Then again, it is important to understand which factors have influenced the poor properties of the industrial samples, since these should have the same layer thicknesses as the tested samples as it was specified in Section 2.1.
Comments on the Quality of English LanguageRemarks included in the "Comments and suggestions for Authors" section.
Author Response
Dear Reviewer, thank you for comments. We have corrected our article according to them. Only comment about the control limits needs explanation . Both measured parameters are between control lines in case of normal distribution analyses. Dispersion of results is the effect of not optimized production process, what is included in conclusions. Optimalisation of production process needs high financial support and engagement of the manufacturer. We are during searching for interested industrial partner.
Reviewer 3 Report
Comments and Suggestions for Authors
1. Please add the main experimental results in the abstract.
2. In the introduction, it is necessary to add the existing research regarding how veneer thickness impacts the performance of flooring (or plywood).
3. How many groups of industrial products were used as controls in this study? Additionally, the model, manufacturer, and detailed composition of these industrial products (e.g., veneer type, number of layers, and thickness of each layer) must be clearly described.
4. In addition to its excellent mechanical properties, plywood is known for its superior dimensional stability. Therefore, the impact of increasing veneer thickness and reducing the number of layers on the dimensional stability of plywood needs further evaluation.
Comments on the Quality of English LanguageMinor editing of English language required.
Author Response
Dear Reviewer, thank you for comments. We have corrected our article according to them. Only point 2 we have not evaluated, because there is not really the existing research regarding veneer thickness in flooring or plywood application. Application of veneers in research and production is based on habits and/or practice.
Round 2
Reviewer 1 Report
Comments and Suggestions for Authors
As I have checked the authors have not implemented all the proposed changes in the revised version of manuscript towards the improvement of their work. Some changes indeed have been implemented, but in my opinion, the manuscript is not yet adequately well-prepared and organized
enough to be accepted for publication in this journal. Please check again one by one the comments and recommendations of mine, in order to apply them in the manuscript text. I remain at your disposal for any clarification.
Here ther has been an improvement. Therefore, acceptable.
Author Response
Authors answer
Dear Reviewer, the English was corrected by native speaker, Mr Nick Faulkner, owner of FLC – Faulkner Language Consultants
As I have checked the authors have not implemented all the proposed changes in the revised version of manuscript towards the improvement of their work. Some changes indeed have been implemented, but in my opinion, the manuscript is not yet adequately well-prepared and organized
enough to be accepted for publication in this journal. Please check again one by one the comments and recommendations of mine, in order to apply them in the manuscript text. I remain at your disposal for any clarification.
We have carefully verified our article according to your’s comments:
Pay attention to the sub-/super-scripts that are not in the appropriate format.
This elements we discuss with Editor
The scientific species names should be in italics.
All wood scientific names are in italic.
In line 67, please provide here a reference.
We have done it.
In line 99, please provide more information.
Additional information is added.
In the case of standard EN 13226:2004, please check it again, since I believe it has been withdrawn and check all the standards for their year of publishing or updated.
We have corrected it.
please add the relevant work of https://link.springer.com/article/10.1007/s00107-024-02114-x#citeas:~:text=DOI-,https%3A//doi.org/10.1007/s00107%2D024%2D02114%2Dx,-Share%20this%20article as a reference.
We have added it.
You should clarify on the type of veneers used (rotary or sliced) from the first time mentioned in materials-methods section.
We have clarified that veneers were rotary cut.
Provide more information concerning the raw materials/elements used in the panel production. which was the equipment used for the rotation cut?
We have added the name of company, because the company do not allowed us to specify their equipment.
The statistical analysis provided is not very clearly presented in the respective section, please provide more information if possible.
The analyses is updated in section 3. Results and discussion
In conclusions section start, something is missing. Chech the first sentence. In line 570, please replace with the symbol %.
We have done it.
Best regards,
Authors
Reviewer 2 Report
Comments and Suggestions for Authors
Dear Authors,
Thank you for addressing my comments.
With reference to Table 3 (line 352 of revised manuscript), please carefully check all items: you have corrected some stiffness values for Group 1, but all other groups have the same issues mentioned previously: the stiffness values are not always proportional to the elastic moduli. If this is just an issue of ranking values in the table, it can be easily solved.
With regard to the comment related to control limits, what was meant was related to the comparison between the Shewhart control charts of Figure 15 (line 587 of revised manuscript) and those of Figure 13 (line 538 of revised manuscript). They both refer to the same groups; however, elastic moduli are often outside LCL and UCL, while the stiffnesses fall only in few cases outside LCL and UCL. This is why I was suggesting an additional explanation, since the two quantities must be proportional, as stiffness is just the modulus of elasticity multiplied by the moment of inertia of the sample. This aspect can be elaborated prior to final acceptance.
Author Response
Authors answer
With reference to Table 3 (line 352 of revised manuscript), please carefully check all items: you have corrected some stiffness values for Group 1, but all other groups have the same issues mentioned previously: the stiffness values are not always proportional to the elastic moduli. If this is just an issue of ranking values in the table, it can be easily solved.
We have done sorting mistakes. It is corrected in Tables and Figures.
With regard to the comment related to control limits, what was meant was related to the comparison between the Shewhart control charts of Figure 15 (line 587 of revised manuscript) and those of Figure 13 (line 538 of revised manuscript). They both refer to the same groups; however, elastic moduli are often outside LCL and UCL, while the stiffnesses fall only in few cases outside LCL and UCL. This is why I was suggesting an additional explanation, since the two quantities must be proportional, as stiffness is just the modulus of elasticity multiplied by the moment of inertia of the sample. This aspect can be elaborated prior to final acceptance.
The explanation about Young’s modulus and stiffness is also developed in the results section.
Best regards,
Authors
Reviewer 3 Report
Comments and Suggestions for Authors
Please summarize the findings in abstract and conclusions of the paper.
Comments on the Quality of English LanguageMinor editing of English language required.
Author Response
Dear Reviewer,
Please summarize the findings in abstract and conclusions of the paper.
We supplemented the abstract with basic research conclusions and changed the conclusions in line with the publisher's suggestions.
The article was corrected by English native speaker- Mr Nick Faulkner, owner of the company FLC - Faulkner Language Consultants.
The Authors would like to thank for insightful review, which will help in further publications.
Round 3
Reviewer 1 Report
Comments and Suggestions for Authors
As I have checked the authors have implemented the proposed changes in the revised version of manuscript towards the improvement of their work. Almost all the changes have been implemented and in my opinion, the manuscript is well-prepared and organized enough to be accepted for publication in this journal. I remain at your disposal for any clarification.
Comments on the Quality of English LanguageAcceptable
Author Response
Dear Reviewer,
The article was corrected by English native speaker- Mr Nick Faulkner, owner of the company FLC - Faulkner Language Consultants.
The Authors would like to thank for insightful review, which will help in further publications.
Reviewer 2 Report
Comments and Suggestions for Authors
Dear Authors,
Thank you for addressing my comments. The manuscript can be accepted for publication.
Author Response
Dear Reviewer,
The Authors would like to thank for insightful review, which will help in further publications.